# Unraveling and Targeting Myocardial Regeneration Deficit in Diabetes

**DOI:** 10.3390/antiox11020208

**Published:** 2022-01-22

**Authors:** Claudia Molinaro, Luca Salerno, Fabiola Marino, Mariangela Scalise, Nadia Salerno, Loredana Pagano, Antonella De Angelis, Eleonora Cianflone, Daniele Torella, Konrad Urbanek

**Affiliations:** 1Department of Medical and Surgical Sciences, University Magna Græcia of Catanzaro, 88100 Catanzaro, Italy; c.molinaro@unicz.it (C.M.); nadia.salerno@unicz.it (N.S.); cianflone@unicz.it (E.C.); 2Department of Experimental and Clinical Medicine, University Magna Græcia of Catanzaro, 88100 Catanzaro, Italy; l.salerno@unicz.it (L.S.); marino@unicz.it (F.M.); m.scalise@unicz.it (M.S.); loredana.pagano@studenti.unicz.it (L.P.); 3Department of Experimental Medicine, University of Campania “L. Vanvitelli”, 80138 Naples, Italy; antonella.deangelis@unicampania.it

**Keywords:** diabetes mellitus, diabetic cardiomyopathy, senescence, adult stem cells, cardiac stem cells

## Abstract

Cardiomyopathy is a common complication in diabetic patients. Ventricular dysfunction without coronary atherosclerosis and hypertension is driven by hyperglycemia, hyperinsulinemia and impaired insulin signaling. Cardiomyocyte death, hypertrophy, fibrosis, and cell signaling defects underlie cardiomyopathy. Notably, detrimental effects of the diabetic milieu are not limited to cardiomyocytes and vascular cells. The diabetic heart acquires a senescent phenotype and also suffers from altered cellular homeostasis and the insufficient replacement of dying cells. Chronic inflammation, oxidative stress, and metabolic dysregulation damage the population of endogenous cardiac stem cells, which contribute to myocardial cell turnover and repair after injury. Therefore, deficient myocardial repair and the progressive senescence and dysfunction of stem cells in the diabetic heart can represent potential therapeutic targets. While our knowledge of the effects of diabetes on stem cells is growing, several strategies to preserve, activate or restore cardiac stem cell compartments await to be tested in diabetic cardiomyopathy.

## 1. Introduction

Diabetes mellitus (DM) is a metabolic disease characterized by hyperglycemia with impaired metabolism of carbohydrates, fats, and proteins because of insulin resistance and/or defects in insulin secretion [1]. The global prevalence of diabetes (particularly type 2 diabetes, T2DM) has been progressively increasing in a troubling manner. The most recent International Diabetes Federation Diabetes Atlas estimates that across the globe more than 450 million adults suffer from diabetes [2]. Diabetes leads to several complications, such as diabetic cardiomyopathy, diabetic nephropathy, diabetic neuropathy, diabetic retinopathy, and diabetic wounding. Cardiovascular disease (CVD) is the leading cause of morbidity and mortality in diabetic patients, accounting for an estimated 80% of all diabetic deaths [3]. The prevalence of heart failure (HF) in diabetic patients ranges from 19% to 26% [4,5,6].

Diabetic cardiomyopathy is defined as a clinical condition where ventricular dysfunction occurs without coronary atherosclerosis and hypertension in patients with DM [5,7]. At the beginning, diabetic cardiomyopathy passes a subclinical period characterized by asymptomatic structural and functional abnormalities, including left ventricular (LV) hypertrophy, cell loss, fibrosis, and cell signaling anomalies. Increased myocardial stiffness, diastolic dysfunction with abnormal filling and elevated LV end-diastolic pressure may progress to systolic abnormalities and LV dilation. With time, diabetic cardiomyopathy can evolve into an overt HF syndrome. In this review, we show that among other pathological phenomena, the diabetic heart suffers from altered cellular homeostasis and deficient myocardial repair indicating the intrinsic regenerative potential of the myocardium as a valid target in diabetic cardiomyopathy.

## 2. Myocardial Pathophysiology in Diabetic Cardiomyopathy

Hyperglycemia, insulin resistance, hyperinsulinemia, and impaired metabolic signaling of insulin are all involved in the pathogenesis of diabetic cardiomyopathy and induce a series of pathophysiological phenomena. Among other things, the increased levels of free fatty acids (FFAs), systemic and tissue inflammation, oxidative stress, and activation of the renin-angiotensin-aldosterone system (RAAS), together with abnormalities in the expression of contractile and regulatory proteins are the most studied factors that drive the progression of the disease. While the major molecular and cellular events involved in the dysregulation of cell homeostasis in the myocardium will be synthetically outlined, this is to underline the fact that there is no population of myocardial cells that can escape diabetic damage.

### 2.1. Inflammation

Chronic inflammation is associated with obesity and aging and seems to contribute to the development of insulin resistance [8,9]. Also, a maladaptive inflammatory response is implicated in the development of diabetic cardiomyopathy [10]. Activation and expression of numerous proinflammatory cytokines, such as tumor necrosis factor α (TNF-α), interleukins 6 (IL-6) and -8 (IL-8), monocyte chemotactic protein1 (MPC-1), intercellular adhesion molecule-1 (ICAM-1), and vascular cell adhesion molecule-1 (VCAM-1) and others, contribute to long-lasting systemic and local low-grade inflammation and sustained cardiac oxidative stress that, in turn, manifests as myocardial disease [11,12,13]. Macrophage migration inhibitory factor (MIF), associated with impaired glucose tolerance [14], is increased in diabetic patients with LV diastolic dysfunction [15]. Proinflammatory responses occur in different cardiac cells, including coronary endothelial and smooth muscle cells, fibroblasts, and cardiomyocytes.

### 2.2. Neurohormonal Activation

The two major HF-related neurohormonal systems, RAAS and sympathetic nervous system (SNS), are also involved in diabetic cardiomyopathy. Serum angiotensin II (AT II) levels are significantly correlated with postprandial glucose concentrations in insulin resistance and T2DM [16]. In the early stages of diabetes, the proinflammatory angiotensin II receptor 1 (AT1R) is upregulated and the anti-inflammatory AT2R is downregulated [17,18]. AT2R inhibition, with a synthetic antagonist PD123319, impairs insulin signaling in healthy C57BL/6 mice [19]. Enhanced AT1R and mineralocorticoid receptor (MR) activation, with the increase in the proinflammatory M1 phenotype in the myocardium [10], increases coronary artery endothelial leukocyte/monocyte adhesion, proinflammatory cytokine expression, and macrophage infiltration and polarization. High plasma aldosterone and overexpression of MR are associated with systemic insulin resistance, hyperglycemia, and dyslipidemia [20]. Large randomized controlled trials have shown that inhibition of the aldosterone/MR signaling pathway reduces morbidity and mortality in diabetic patients with both mild and moderately severe HF [21]. Elevated SNS activity in a diabetic heart is the main feature of cardiac autonomic neuropathy, a secondary complication of sustained hyperglycemia. Through an enhanced β-1 adrenergic receptor signaling, SNS activity promotes cardiac hypertrophy, interstitial fibrosis, cardiomyocyte apoptosis, and abnormalities in heart rate control and vascular hemodynamics [22]. Finally, SNS together with RAAS increase advanced glycation end products (AGEs) [23].

### 2.3. Nitric Oxide (NO) and Reactive Oxygen Species (ROS)

Impaired insulin signaling negatively affects NO production, both directly and through increased ROS levels. Under physiological conditions, insulin regulates nutrient supply by stimulating NO production and consequent vasodilation [24], but in T2DM, altered insulin metabolic signaling reduces insulin-mediated coronary endothelial nitric oxide synthase (eNOS) activity and NO production [25] that, associated with the altered cytoplasmic Ca^2+^ concentration, causes excessive coronary vasoconstriction. In patients with T2DM, endothelium-dependent vasodilation is reduced [26]. Hyperglycemia and insulin resistance also increase ROS production. ROS react with circulating NO leading to a reduction of bioavailable NO [27]. Moreover, elevated ROS increases polyol pathway flux, formation of AGEs, expression of the receptor for AGEs, PKC signaling and the hexosamine pathway [28]. In T2DM there is an increase of FFA release from adipose tissue which are used by the myocardium as the main source of energy, due to impaired glucose uptake [29]. With the same mechanism of hyperglycemia and insulin resistance, FFA leads to the increased production of ROS. It has been demonstrated that palmitate (a major saturated FFA), as well as a high glucose level, stimulates ROS production through PKC-dependent activation of NAD(P)H oxidase in cultured vascular cells [30].

### 2.4. Endothelial Dysfunction

Although the definition of diabetic cardiomyopathy includes the absence of coronary disease, it may be associated with coronary microvascular dysfunction that impairs coronary blood flow and myocardial perfusion [31,32]. Coronary microcirculation presents structural and functional abnormalities. Lumen obstruction, inflammatory infiltration, vascular remodeling, and perivascular fibrosis accompany endothelial and smooth muscle cell dysfunction with impairment of vascular relaxation-constriction coupling [33,34]. In the early stages of diabetes, despite defective NO-mediated vasodilation, vascular function is preserved by normal or even enhanced endothelium-derived hyperpolarizing factor that induces vasodilation. NO- and endothelium-derived hyperpolarizing- factor-induced vasodilation are eventually exhausted, leading to significant dysfunction of the microcirculation [35]. Moreover, persistently elevated plasma endothelin-1 levels, reduced eNOS activity and NO production are linked to the development of cardiac fibrosis and diastolic dysfunction in diabetic patients and diabetes-induced cardiac fibrosis is prevented by endothelial cell-specific endothelin-1 knockout [36,37,38].

### 2.5. Glucose Transport and Calcium Homeostasis

Like skeletal muscle, liver, and adipose tissue, the transport of glucose into cardiac cells is mediated by glucose transporter type 4 (GLUT4). Insulin stimulates the translocation of GLUT4 to the cell membrane and the subsequent absorption of glucose. Impaired insulin signaling reduces GLUT4 recruitment to the plasma membrane and glucose uptake, leading indirectly to decreased activity of the sarcoplasmic reticulum (SR) Ca^2+^ pump (SERCA pump), which needs ATP to translocate Ca^2+^ from the cytoplasm to SR. Following the excitation-contraction events in the cardiomyocyte, the removal of intracellular Ca^2+^ by the SERCA pump, together with the Na/Ca^2+^ exchanger (NCX) and the Ca^2+^ ATPase pump (PMCA) present on the plasma membrane, allows cell relaxation. In DCM, the altered concentration of cytoplasmic Ca^2+^ changes the functioning of all these transporters resulting in increased action potential duration and prolonged relaxation time [39].

### 2.6. Myocardial Fibrosis

Myocardial fibrosis, which increases myocardial stiffness, involves the deposition of interstitial collagen type I and III and its cross-linking, perivascular fibrosis, the thickening of the basement membrane in small coronary vessels, coronary microvascular sclerosis and microaneurysms [40]. Impaired passive relaxation also results from crosslinking of the extracellular matrix that is promoted by AGEs. Moreover, interaction of AGEs with receptors for AGE (RAGEs) on cardiomyocyte surfaces induces maladaptive proinflammatory responses that further increases matrix synthesis and connective tissue accumulation [23].

### 2.7. Myocardial Cell Loss

Diabetic cardiomyopathy is characterized by endoplasmic reticulum stress where oxidative stress, lipotoxicity, inflammation and the accumulation of misfolded proteins impair the function of the cardiac cell endoplasmic reticulum. This results in the inhibition of protein synthesis and degradation of misfolded or damaged proteins and, ultimately, the increase in apoptosis. Also, altered mitochondrial Ca^2+^ handling promotes mitochondrial respiratory dysfunction leading to cell death. Metabolic stress-induced mitochondrial dysfunction also increases the Ca^2+^ overload opening of mitochondrial permeability transition pores [41,42]. Diabetes is characterized by an 85-fold, 61-fold, and 26-fold increase in apoptosis of myocytes, endothelial cells, and fibroblasts, respectively. Furthermore, in patients with diabetes and hypertension, necrosis is increased by 7-fold in myocytes, 18-fold in endothelial cells, and 6-fold in fibroblasts (Figure 1) [43,44].

### 2.8. Cell Senescence

Cellular senescence is a part of a pathogenic loop in diabetes, as both a cause and a consequence of metabolic changes and tissue damage. Cellular senescence is a dynamic process characterized by a stable and largely irreversible arrest of proliferation due to several triggers, such as DNA damage, accumulation of ROS, telomere shortening and chronologic age [46,47]. The association between diabetes and senescence is complex: likely, the diabetic microenvironment is permissive to the formation and accumulation of senescent cells, and senescent cells may contribute to the tissue dysfunction and comorbidities observed in T2DM. High glucose induces premature senescence in endothelial cells, renal mesangial cells, fat-derived stem cells (also known as preadipocytes or fat cell progenitors) and fibroblasts. The underlying mechanisms are not completely understood, but mitochondrial dysfunction and the production of ROS [48] and/or the formation of AGEs play a leading role [49]. Indeed, one of the senescence-associated secretory phenotype (SASP) factors, HMGB1, an agonistic ligand of RAGEs, link inflammatory signaling to p53-dependent cellular senescence [50]. Cell senescence is also linked to chronic inflammation as SASP may include interleukin IL-6, IL-8 and MCP; the same components that are increased in adults and obese adolescents [51,52]. A high concentration of IL-6 and a combined increase of IL-6 and IL-1b, both SASP factors, are independent predictors of diabetes [53]. In addition, chronic exposure to insulin-like growth factor-1 (IGF-1), can lead to premature senescence via p53 pathway [54]. The activity of one of its effectors, Akt, increases cell senescence, while the inhibition of Akt increases the duration of the replicative life of endothelial cells in vitro [55]. Cellular senescence also has deleterious effects on adipose tissue, leading to a reduced capacity to accumulate fat and to the “spillover” of FFAs, with their effects on the heart [56]. Given the growing importance of senescence in the progression of various chronic diseases, targeting senescent cells emerges as a plausible therapeutic strategy.

## 3. Diabetes and Adult Stem Cell Function

Adult stem cells that reside in tissue-specific niches have high proliferative potential and the capacity to differentiate in all cell types of the organ in which they operate. Adult stem cells are implicated in homeostasis, regeneration, and aging [57]. Diabetes impairs the function of endogenous stem cells contributing to organ dysfunction but the exact mechanisms by which diabetes alters stem cells is not completely understood [58,59]. It involves enhanced oxidative stress, proinflammatory status and mitochondrial dysfunction. Cellular and molecular alterations induced by diabetes lead to impaired differentiation, proliferation, angiogenic potential [60,61] and migration and homing (a condition known as mobilopathy) [62]. An example of stem cells affected by diabetes are bone marrow-derived stem cells (BMSCs). In streptozotocin (STZ)-induced diabetic rats, BMSCs showed significantly decreased proliferative capacity. The production of cytokines such as vascular endothelial growth factor (VEGF) and IGF-1, and myogenic differentiation was decreased in the diabetes group. At the same time, the rate of BMSCs apoptosis was significantly higher [63]. Consistently, the analysis of the bone marrow of patients with T2DM showed a reduction of hematopoietic tissue with reduced number of CD34-positive BM progenitor cells (BMPCs) and increased progenitor cell apoptosis [64]. Diabetes-induced modifications in progenitor cell biology affect their normal behavior during tissue repair. Diabetic patients suffer from exaggerated inflammation of wounds [65], which can be due to the presence of a disproportionate amount of proinflammatory M1 macrophages contributing to impaired wound healing [66]. This shift towards a proinflammatory phenotype seems to be due to changes at the stem cell compartment, in which diabetes prompts a preferential myeloid-lineage commitment in bone marrow progenitors [67].

Diabetes also affects endothelial progenitor cells (EPCs) which are a population of circulating cells capable of differentiating into endothelial cells and make up blood vessels [68]. The impaired EPCs’ function results from diabetes-induced damage to the peripheral circulation and to the bone marrow. Rodents with STZ-induced diabetes show a low level of circulating EPCs and an impaired cell mobilization [69]. Moreover, in diabetic mice there is a lower release of a chemoattractant signaling molecule stromal-derived factor-1 alpha (SDF-1α) from local tissues and a decreased activation of eNOS in the bone marrow [70]. In humans, reduced circulating EPCs is a cardiovascular risk factor [71]. Diabetes-associated changes in EPCs function include defects in proliferation and vascular tube formation in vitro [72]. Diabetes also induces cell mobilopathy resulting in the abnormal retention of stem/progenitor cells in the bone marrow [62]. Diabetes-related retention of stem/progenitor cells in the bone marrow involves altering the dipeptidyl peptidase-4/stromal-derived factor-1 alpha (DPP-4/SDF1-α) axis, which is essential to establish a cytokine gradient towards the peripheral blood. Furthermore, CXCL12/SDF-1α and its cognate receptor CXCR4, known to mediate stem cell homing, are found to be reduced in stem cells (adipose tissue-derived stem cells) from diabetic patients [73,74,75,76].

## 4. Diabetes and Cardiac Stem Cell Biology

The notion that the heart is a terminally differentiated organ has been challenged with evidence of the existence of resident endogenous cardiac stem cells (CSCs). CSCs were first identified by the expression of the stem cell factor (SCF) receptor, c-kit, a type III receptor tyrosine kinase (also called CD117 or SCFR) [74,77]. Although it has been proven that these cells are necessary and sufficient for functional cardiac regeneration and repair [78,79], the ongoing debate regarding the role of CSCs in myocardial homeostasis in repair is fueled by unsuccessful fate mapping studies that were unable to follow cell fate in vivo. Specifically, in the Cre-Knock-in mice, the very low number of CSC-derived cardiomyocytes reflected functional failure of the molecular construct used to track the progeny of c-kit^pos^ CSC [80,81,82,83,84]. Adult cardiac c-kit^pos^ cell population is heterogeneous and c-kit alone is not sufficient to distinguish CSCs from other c-kit expressing cells. Approximatively 90% of c-kit^pos^ cells in the adult heart co-express blood and endothelial cell lineage markers, such as CD45 and CD31, while only 10% are CD45 and CD31 negative (Lin^neg^) [85,86,87]. The sequential sorting strategy of the total cardiac cell population obtained from the adult heart (CD45/CD31 negative followed by c-kit-positive sorting), provides cell population enriched for Lin^neg^c-kit^pos^. Yet these cells are still heterogenous as their ability to self-renew and their multipotentiality is restricted to 10% of the cells. It follows that the identity of a CSC can only be resolved by analyzing a single cell-derived clone [85,88]. A set of membrane and nuclear markers, together with c-kit, outlines the phenotype of a Lin^neg^c-kit^pos^ clone characterized by the expression of PDGFR-α, CD166, SSEA-1, Nestin, CD90, ROR2, CD146, CD3, Bmi-1, Tert, Gata-4 and Nkx2.5, Oct3/4, Nanog, Klf-4 and Sox-2 [85,86]. Adult CSCs are multipotent, and when grown in a specific differentiation medium acquire the phenotype of the three main cardiac cells (cardiomyocytes, smooth muscle and endothelial cells). However, cardiomyocyte derived from clonal CSCs showed a phenotype that more closely resembles neonatal, immature cardiomyocytes compared with the adult, terminally differentiated cardiomyocytes [89]. Contracting cardiomyocyte derived from clonal CSCs have an intermediate transcriptional profile with the levels of mRNA of transcription factors and myofilament proteins that fall between fetal and neonatal cardiomyocytes. Cardiomyocyte derived from CSCs up-regulate the cardiomyo-miRs (miR-1a 3p, miR-133a-1, miR-133a-2, miR-204, miR-335, miR-486, miR-490, and miR-499) to a lower extent than adult cardiomyocytes. However, the specific networks of miRNA/mRNAs operative in CSC-derived cardiomyocytes closely resembled those of adult cardiomyocytes [89].

Several lines of evidence document that diabetes negatively affects CSCs homeostasis leading to insufficient replacement of old, dying cells and the acquisition by the heart of a senescent phenotype [90,91,92]. Diabetes-associated cell senescence and the impact of diabetes-related high ROS levels depend, at least partly, on functional p66Shc. This adaptor protein regulates the ROS-generating system and has also been identified as a sensor of oxidative stress-induced apoptosis [93,94]. Changes in p66Shc expression and/or function may play a role in the pathogenesis of T2DM [95]. Interestingly, the ablation of p66Shc had remarkably beneficial consequences on the viability and function of murine c-kit^pos^ CSCs, positively interfering with death stimuli and inhibition of CSCs growth and differentiation. CSC proliferation and myocyte generation were potentiated by the absence of p66Shc contributing to the prevention of HF development [90]. Several reports have shown that the decrease of the proliferative and differentiation potential of adult CSCs from diabetic patients is related to their senescence phenotype [92,96] and that the changes in chromatin conformation underlie the impaired proliferation, differentiation, and senescence of diabetic CSCs [60,97,98].

The activation of CSCs in diabetic infarcted rats was drastically reduced and accompanied by a worsening of cardiac function. Further, DM reduced myocardial expression of the SCF along with a decrease of phosphorylation of ERK1/2 and p38 MAPK leading to inhibition of CSC mobilization in the peri-infarct zone [99]. DM has been shown to significantly reduce the activity of key enzymes of the pentose phosphate pathway (G6PD and transketolase) in cardiac Sca-1^pos^ cells from diabetic mice, resulting in decreased antioxidant defense mechanisms, accumulation of glucose intermediates and activation of apoptosis. Exposure of non-diabetic murine Sca-1^pos^ cardiac progenitor cells (CPCs) or human CD105^pos^ cardiac progenitors to high glucose produced similar biochemical and functional abnormalities [97]. Hyperglycemia modulates the electrophysiological properties of human CPCs by altering the expression of miRNAs and potassium channels, resulting in impaired potassium flow and in the development of diabetic arrhythmias. High glucose leads to miR-1/133-dependent changes in the electrophysiological properties of human CPCs, including the targeted suppression of KCNE1 and KCNQ1 [100]. The miRNA-mediated changes in electrophysiology of CPCs contribute to a declined repair capacity of these cells repair in diabetic cardiomyopathy [90,101]. The harmful effect of DM on CSCs also involves miR-34a. miR-34a is associated with the pro-survival protein SIRT1 (a histone deacetylase, identified as a critical regulator of cellular longevity) and exerts several effects depending on the cellular context. In the early stage of diabetes, adult cardiac muscle cells and CSCs show increased miR-34a levels, downregulated SIRT1 and a higher pro-apoptotic caspase-3/7 activity. miR-34a inhibition reduced not only high glucose-induced cardiomyocyte apoptosis but also CPCs proliferation [102]. Overall, there is evidence showing that CSC population, crucial for the maintenance of cardiac homeostasis and repair, is a new target of pathophysiological milieu in diabetes.

## 5. Cell Therapy for Diabetic Cardiomyopathy

The potential applications of regenerative medicine can include the use of stem cells also for diabetic cardiomyopathy. In an experimental setting, diabetic rats who received MSC by subcutaneous infusion showed lower blood sugar and increased serum insulin levels. The MSC administration also improved the heart function in diabetic rats [103]. Similarly, improved cardiac function was detected in diabetic animals following intravenous administration of MSCs. Transplanted MSCs were found in the heart and a small percentage expressed cardiac markers troponin T and myosin heavy chain. In diabetic myocardium, treatment with MSCs significantly increased myocardial arteriolar density, decreased the collagen content, increased matrix metalloproteinase-2 (MMP-2) activity and decreased transcription of MMP-9 gene [104]. However, the improved heart function might be due, indirectly, at least in part to the concomitant regeneration of pancreatic cells, observed in both studies [103,104]. Similarly, preconditioned MSCs increased the capillary density and reduced fibrosis in the diabetic myocardium. These benefits were accompanied by increase in MMP-2 activity, inhibition of TGF-β, raise of Bcl-2/Bax ratio and decreased activation of caspase-3 [105]. Such results were not obtained when using diabetic cells. Diabetic bone marrow mononuclear cells were unable to improve post- myocardial infarction cardiac function, as opposed to healthy cells [106].

Given the role of CSCs in the myocardial biology, the intrinsic regenerative potential of the adult heart represents a natural and valid therapeutic target. The preservation and mobilization of CSCs, that were successfully tested in the ischemic and drug-induced cardiomyopathies analogous approach, seems plausible also in a diabetic cardiomyopathy. Considering the mechanisms involved in the deterioration of CSC function in diabetes, several strategies can be envisioned to protect, activate, and counteract cellular senescence.

One of the potential targets is the histone deacetylase SIRT1 that plays a major role in inflammation, apoptosis, and oxidative stress responses. The protective effects of SIRT1 activation were recognized also in CSCs. Several molecules acting on SIRT proteins are in the pipeline, but the most studied so far is a natural polyphenol resveratrol known for its antioxidant and anti-apoptotic properties. Resveratrol, through the activation of SIRT1, interferes with the damage of CSCs induced by the cardiotoxic drug doxorubicin [107]. Specifically, SIRT1 activation in human CSCs reduced acetylation of p53, enhanced defense against oxidative stress and prevented cell senescence. Improved cell survival, growth and functional properties translated into partial preservation of the cardiac function [108,109,110]. Also, in diabetic rats, resveratrol reduced CSC loss and ameliorated cardiac function by reducing inflammatory state and pathological ventricular remodeling [111]. At cardiomyocyte level, resveratrol reduced the incidence of cell death and enhanced the expression of sarcoplasmic/endoplasmic reticulum calcium ATPase 2a (SERCA2a) in experimental models of diabetes [112,113].

The activation of CSCs has also been achieved with statins, a class of drugs used to lower cholesterol and to reduce cardiovascular risk. In addition, statins may exert protective cardiovascular effects independently of cholesterol lowering by so-called pleiotropic effects. In a recent study, statins, by sustaining Akt activation, promoted CSC growth and differentiation in vitro and in vivo. Rosuvastatin, simvastatin and pravastatin increased CSC proliferation, clonal expansion and cardiosphere formation. While these effects allowed an improved myocardial remodeling after coronary occlusion in rodents, it is not known whether similar effects might contribute to the beneficial effects in patients [114].

The regenerative potential of the heart can also be preserved with interventions that revert a senescence process that is a dynamic, rather than a terminal, phenomenon [115,116]. In human cardiac progenitors, the knockdown of p16INK4A, a central player in cellular senescence, reverses the senescent phenotype, increasing cell proliferation and survival capacity [117]. Interestingly, the use of senolytic drugs for the selective clearance of senescent cells from “aged” tissues may be useful for treating cardiac deterioration and rejuvenating the regenerative capacity of the heart. Dasatinib (an FDA-approved tyrosine kinase inhibitor) and quercetin (a flavonoid present in fruits and vegetables) preserve cycling-competent CSCs, while senescent CSCs are induced to selective apoptosis [118]. Such an approach is yet to be tested in diabetic cardiomyopathy.

## 6. Conclusions

The multifaceted nature of the molecular and cellular mechanisms underlying diabetic cardiomyopathy is a challenge for basic and clinical researchers. As “the magic bullet” most likely will not be found, it is reasonable to envision an approach that would combine the anti-inflammatory, cell-protective, senolytic-senostatic and cell-activating strategies. Thus, the advanced possibility of preserving and restoring the intrinsic, stem cell-mediated, regenerative potential of the diabetic heart carries a significant translational potential. To this regard, precision medicine, which has the goal of providing “the right treatment to the right patient at the right time” [119], having a significant potential to play an important role in the treatment of cardiovascular diseases [120], by taking into consideration the individual differences in genetic, and environmental and life style factors in determining a person’s disease phenotype, could have a significant impact on developing these new specific therapies accounting for the multifaced nature of diabetes-induced cardiovascular disease, and particularly DCM.

## Figures and Tables

**Figure 1 antioxidants-11-00208-f001:**
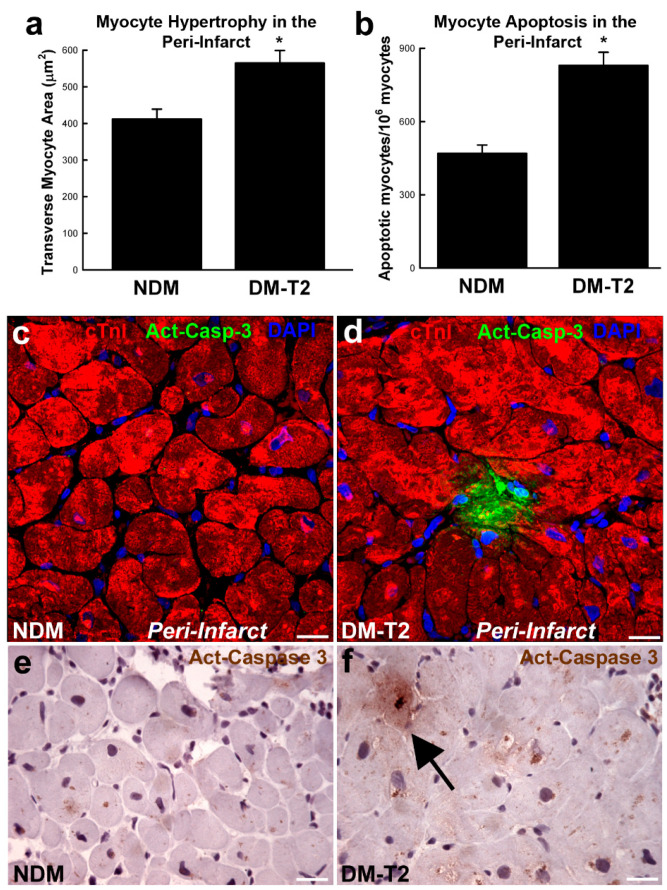
(**a**,**b**) Bar graphs showing cardiomyocyte size and apoptosis in peri-infarct areas samples from patients with type 2 diabetes mellitus (DM-T2) compared with non-diabetic patients (NDM); * *p* < 0.01 vs. NDM, n = 20 per group. (**c**–**f**) Representative confocal microscopy and DAB-staining light microscopy images showing an activated caspase-3 (Act-Casp-3)–positive apoptotic myocyte in the peri-infarct zone of a DM-T2 sample compared with NDM. Quantitative data are expressed as mean ± SE. DAB indicates 3,3′-diaminobenzidine; NHE-1, sodium hydrogen exchanger-1. Adapted from [45].

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
