# Peer review of "Unraveling and Targeting Myocardial Regeneration Deficit in Diabetes"

_antioxidants, 2022, doi:10.3390/antiox11020208_

Round 1

Reviewer 1 Report

Diabetes mellitus (DM) is a major public health problem. Diabetic cardiomyopathy (DCM) typically refers to structural and functional abnormalities of the myocardium in DM patients with without coronary artery disease, hypertension or valvular heart disease. Diabetic cardiomyopathy can lead to heart failure (HF), which can occur in both type 1 DM (T1D) and type 2 DM (T2D). In this manuscript, the authors provide a comprehensive review of major aspects of the pathogenesis of diabetic cardiomyopathy (DCM), encompassing inflammation, neurohormonal activation, oxidative stress, endothelial dysfunction, impairment of calcium homeostasis, myocardial fibrosis and cell loss, cell senescence, cardiac stem cell (CSC) biology, and stem cell therapy for DCM.

(I) Major Comments

Diabetes mellitus (DM) is a metabolic disease characterized by hyperglycemia. With the improvement of living standard, the incidence of DM continues to rise across the world (e.g., American Diabetes Association. Diagnosis and classification of diabetes mellitus. Diabetes Care. 2012;35 Suppl 1:S64-71. PMID: 22187472). DM causes long-term damage to multiple organs, including the heart. Diabetic cardiomyopathy (DCM) is a common and severe complication of DM and increases the risk of heart failure with heavy social and economic burden worldwide. DCM contributes to the higher incidence of heart failure in patients with diabetes (e.g., Aneja A, Tang WH, Bansilal S, Garcia MJ, Farkouh ME. Diabetic cardiomyopathy: insights into pathogenesis, diagnostic challenges, and therapeutic options. Am J Med. 2008;121:748-57. PMID: 18724960.). In this manuscript, the authors appear to have provided a well-organized, multi-faceted and logical review of molecular mechanisms of DCM, and have indicated that stem cell therapy can provide myocardial repair and regeneration for treatment of cardiac diseases.

I have the following major comments.

(1) Page 1, lines 20-22, the authors stated that

"Oxidative stress, chronic inflammation and metabolic dysregulation damage the population of endogenous cardiac stem cells contribute to myocardial cell turnover and repair after injury."

However, the in the main text, the authors introduced "2.1 Inflammation" before "2.3. Nitric oxide (NO) and reactive oxygen species (ROS)", and therefore,

the above statement could be corrected to

"Chronic inflammation, oxidative stress, and metabolic dysregulation damage the population of endogenous cardiac stem cells, which contribute to myocardial cell turnover and repair after injury."

(2) Page 4, Figure 1 legend, Page 4, line 155,

"DMT2"

could be corrected to

"DM-T2”

(3) Page 4, Figure 1 legend, Page 4, line 157,

"NHE-1, sodium-hydrogen exchanger-1"

could be corrected to

"NHE-1, sodium hydrogen exchanger-1"

Although Reference34 from which this figure is adapted from did state "sodium-hydrogen exchanger-1", however, the more standard term for the abbreviated term "NHE-1" is "sodium hydrogen exchanger-1".

(4) Page 4, lines 147-148,

"interaction of AGEs with their receptors (RAGE) on cardiomyocyte surfaces induces"

could be corrected to

"interaction of AGEs with receptors for AGE (RAGEs) on cardiomyocyte surfaces induces"

(5) Page 6, lines 233-234,

"Furthermore, CCXCR4 and CXCL12/SD1F, chemokines known to mediate stem cell homing, are found to be reduced"

could be corrected to

"Furthermore, CXCL12/SDF-1α and its cognate receptor CXCR4, known to mediate stem cell homing, are found to be reduced"

(6) Page 6, lines 239-240,

"the expression of the stem cell factor receptor, c-kit, a type III re-ceptor tyrosine kinase (also called CD117 or SCF-R)"

could be corrected to

"the expression of the stem cell factor (SCF) receptor, c-kit, a type III re-ceptor tyrosine kinase (also called CD117 or SCFR)"

(7) Page 6, lines 260-264,

"CMs derived from clonal CSCs showed a phenotype that more closely resemble neonatal, immature cardiomyocytes compared with the adult, terminally differ-entiated CMs [76]. Contracting CMs derived from clonal CSCs have an intermediate transcriptional profile with the levels of mRNA of transcription factors and myofilament proteins that fall between fetal and neonatal CMs."

could be corrected to

"Cardiomyocyte derived from clonal CSCs showed a phenotype that more closely resemble neonatal, immature cardiomyocytes compared with the adult, terminally differentiated cardiomyocytes [76]. Contracting cardiomyocyte derived from clonal CSCs have an intermediate transcriptional profile with the levels of mRNA of transcription factors and myofilament proteins that fall between fetal and neonatal cardiomyocytes."

(8) Page 6, lines 264-268,

"Cardiomyocytes derived from CSCs up-regulate the cardiomyo-miRs (miR-1a 3p, miR-133a-1, miR-133a-2, miR-204, miR-335, miR-486, miR-490, and miR-499) to the lower extent than adult CMs. However, the specific networks of miRNA/mRNAs operative in CSCs-derived CMs closely resembled those of adult CMs [76]."

could be corrected to

"Cardiomyocyte derived from CSCs up-regulate the cardiomyo-miRs (miR-1a 3p, miR-133a-1, miR-133a-2, miR-204, miR-335, miR-486, miR-490, and miR-499) to the lower extent than adult cardiomyocytes. However, the specific networks of miRNA/mRNAs operative in CSC-derived cardiomyocytes closely resembled those of adult cardiomyocytes [76]."

(9) Page 6, lines 295-297, the authors stated that

"High glucose leads to miR-1/133-dependent changes in the electrophysiological properties of human CPCs, including the targeted suppression of KCNE1 and KCNQ1 [84]"

The authors are referring to

Reference84, i.e.,

Li Y, Shelat H, Geng YJ. IGF-1 prevents oxidative stress induced-apoptosis in induced pluripotent stem cells which is mediated by microRNA-1. Biochem Biophys Res Commun. 2012;426:615-9. PMID: 22982320,

and in the main text of this article, the authors did not mention miR-133, KCNE1, or KCNQ1.

Therefore, the above reference, Reference84,

could be corrected to

Li Y, Yang CM, Xi Y, et al., MicroRNA-1/133 targeted dysfunction of potassium channels KCNE1 and KCNQ1 in human cardiac progenitor cells with simulated hyperglycemia. Int J Cardiol. 2013;167:1076-8. PMID: 23157812.

(10) Pages 8-9, in "6. Conclusions" section,

Precision medicine, which has the goal of providing "the right treatment to the right patient at the right time" (as stated by Currie G, Delles C. Precision Medicine and Personalized Medicine in Cardiovascular Disease. Adv Exp Med Biol. 2018;1065:589-605. PMID: 30051409), has emerged to play an important role in treatment of cardiovascular medicine (e.g., Leopold JA, Loscalzo J. Emerging Role of Precision Medicine in Cardiovascular Disease. Circ Res. 2018;122:1302-1315. PMID: 29700074,), and the authors shall provide a discussion on how precision medicine, which takes into consideration of individual differences in genetic, and environmental and life style factors in determining a person’s disease phenotype, could have a significant impact on developing new therapies for diabetes-induced cardiovascular disease, particularly DCM.

(II) Minor Comments

The authors shall perform a thorough and careful English editing, and the following grammatical and typographical errors that should be corrected:

Page 1, line 37,

"diabetic cardiomyopathy (DCM)"

could be corrected to

"diabetic cardiomyopathy"

Page 1, line 40,

"The prevalence of heart failure"

could be corrected to

"The prevalence of heart failure (HF)"

Page 2, line 49,

"can evolve into overt heart failure syndrome"

could be corrected to

"can evolve into overt HF syndrome"

Page 2, line 57,

"free fatty acids (FFA)"

could be corrected to

"free fatty acids (FFAs)"

Page 2, line 69,

"interleukins 6 (IL-6) and 8 (IL-8), monocyte chemotactic protein 1 (MPC-1)"

could be corrected to

"interleukins-6 (IL-6) and -8 (IL-8), monocyte chemotactic protein-1 (MCP-1)"

Page 2, lines 69-71,

"ad-hesion molecule intercellular 1 (ICAM-1), and vascular cell adhesion molecule 1 (VCAM-1)"

could be corrected to

"intercellular adhesion molecule-1 (ICAM-1), and vascular cell adhesion molecule-1 (VCAM-1)"

Page 2, line 74,

"with left ventricular diastolic dysfunction"

could be corrected to

"with LV diastolic dysfunction"

Page 2, line 78,

"The two major heart failure-related neurohormonal"

could be corrected to

"The two major HF-related neurohormonal"

Page 2, line 79,

"are also involved in DCM"

could be corrected to

"are also involved in diabetic cardiomyopathy"

Page 2, lines 81-82,

"angiotensin II receptor 1 (AT-1R) is upregulated and the anti-inflammatory AT-2R is downregulated [10]. AT-2R"

could be corrected to

"angiotensin II receptor 1 (AT1R) is upregulated and the anti-inflammatory AT2R is downregulated [10]. AT2R"

such that "AT-1R" is corrected to "AT1R", and "AT-2R" is corrected to "AT2R"

Page 2, line 84,

"Enhanced AT-1R and mineralocorticoid receptors (MR)"

could be corrected to

"Enhanced AT1R and mineralocorticoid receptor (MR)"

Page 2, line 91,

"moderately severe heart failure"

could be corrected to

"moderately severe HF"

Page 3, line 97,

"2.3. NO and ROS"

could be corrected to

"2.3. Nitric oxide (NO) and reactive oxygen species (ROS)"

Page 3, line 98,

"affects nitric oxide (NO) production"

could be corrected to

"affects NO production"

Page 3, lines 110-111,

"FFA lead to the increased"

could be corrected to

"FFA leads to the increased"

Page 5, line 194,

"FFA, with their effects on the heart"

could be corrected to

"FFAs, with their effects on the heart"

Page 5, line 199,

"to differentiate in all the cell types of the organ in which"

could be corrected to

"to differentiate into all cell types of the organ in which"

Page 6, line 219,

"in bone marrow progenitors[58]"

could be corrected to

"in bone marrow progenitors [58]"

Page 6, lines 225-226,

"signaling molecule stromal-derived factor 1a (SDF-1a)"

could be corrected to

"signaling molecule stromal-derived factor-1 alpha (SDF-1a)"

In above corrected "SDF-1a", "a" denotes Greek symbol alpha

Page 6, lines 231-232,

"the dipeptidyl peptidase-4/stromal-derived factor-1 alpha (DPP4/SDF1-α)"

could be corrected to

"the dipeptidyl peptidase-4/stromal-derived factor-1 alpha (DPP-4/SDF-1α)"

Page 6, line 238,

"existence of a resident endogenous cardiac stem cell (CSCs)"

could be corrected to

"existence of resident endogenous cardiac stem cells (CSCs)"

Page 6, line 241,

"regeneration and repair[66]"

could be corrected to

"regeneration and repair [66]"

Page 6, line 244,

"in the Cre-Knock-In mice"

could be corrected to

"in the Cre-knock-in mice"

Page 7, line 276,

"remarkable beneficial consequences"

could be corrected to

"remarkably beneficial consequences"

Page 7, line 279,

"contributing to prevent the development of HF [77]"

could be corrected to

"contributing to the prevention of the development of heart failure [77]"

Page 7, line 291,

"CPCs or human"

could be corrected to

"cardiac progenitor cells (CPCs) or human"

Page 7, lines 292-293,

"Hyperglycemia modulates also the electrophysiological properties of human CPCs"

could be corrected to

"Hyperglycemia also modulates the electrophysiological properties of human CPCs"

Page 7, line 301,

"as critical regulator of cellular longevity"

could be corrected to

"as a critical regulator of cellular longevity"

Page 7, lines 304-305,

"miR-34a inhibition, reduced high glucose-induced cardiomyocyte apoptosis but also reduced CPCs proliferation [86]."

could be corrected to

"miR-34a inhibition reduced not only high glucose-induced cardiomyocyte apoptosis but also CPCs proliferation [86]."

Page 7, line 312,

"also improved the heart function in diabetics rats [87]"

could be corrected to

"also improved the heart function in diabetic rats [87]"

Page 8, lines 323-324,

"Diabetic BM mononuclear cells were unable to improve post-MI cardiac function"

could be corrected to

"Diabetic bone marrow mononuclear cells were unable to improve post-myocardial infarction cardiac function"

Page 8, line 338,

"interferes with the damage that of CSCs induced by"

could be corrected to

"interferes with the damage of CSCs induced by"

Page 8, line 342,

"Also in diabetics rats, resveratrol"

could be corrected to

"Also in diabetic rats, resveratrol"

Page 8, line 345,

"of (SERCA2a) in experimental models of diabetes"

could be corrected to

"of sarcoplasmic/endoplasmic reticulum calcium ATPase 2a (SERCA2a) in experimental models of diabetes"

Page 8, lines 347-348,

"Statins may exert cardiovascular pro-tective effects also independently of cholesterol lowering"

could be corrected to

"In addition, statins may exert protective cardiovascular effects independently of cholesterol lowering"

Page 8, line 355,

"revert senescence process"

could be corrected to

"reverse senescence process"

Page 8, lines 362-363,

"Such approach is yet to be tested"

could be corrected to

"Such an approach is yet to be tested"

Reviewer 2 Report

This review article about DM cardiomyopathy is well written with good description about this issue in details. I have no further comments about this. 

Author Response

Please, see the attached file

Reviewer 3 Report

Review focused on an actual and interesting topic. However, there are several points which authors should explain and clarify.

I have the following comments:

1. Graphs and figures presented in Figure 1 are not directly related to discussed topic – see content of the sentence in lines 143-146 and sentence in lines 176-179:

“Myocardial fibrosis, that increases myocardial stiffness involves the deposition of interstitial collagen type I and III and its cross-linking, perivascular fibrosis, the thickening of the basement membrane in small coronary vessels, coronary microvascular sclerosis and microaneurysms (Figure 1).” 

“The association between diabetes and senescence is complex: likely, the diabetic microenvironment is permissive to the formation and accumulation of senescent cells, and senescent cells may contribute to the tissue dysfunction and comorbidities observed in T2D (Figure 1).”

2. In review are in several cases cited not original experimental studies but only other review articles. And I suppose that your manuscript is not only review of review articles. Some examples:

Lines 67-72: “Activation and expression of numerous proinflammatory cytokines, such as tumor necrosis factor α (TNF-α), interleukins 6 (IL-6) and 8 (IL-8), monocyte chemotactic protein 1 (MPC-1), adhesion molecule intercellular 1 (ICAM-1), and vascular cell adhesion molecule 1 (VCAM-1) and others, contribute to long-lasting systemic and local low-grade inflammation and sustained cardiac oxidative stress that, in turn manifests as myocardial disease [10].”

Lines 81-82:”In early stages of diabetes, the proinflammatory angiotensin II receptor 1 (AT-1R) is upregulated and the anti-inflammatory AT-2R is downregulated [10].”

Lines 104-106:” Hyperglycemia and insulin resistance also increase ROS production. ROS react with circulating NO leading to a reduction of bioavailable NO [21].”

Lines 143-146:” Myocardial fibrosis, that increases myocardial stiffness involves the deposition of interstitial collagen type I and III and its cross-linking, perivascular fibrosis, the thickening of the basement membrane in small coronary vessels, coronary microvascular sclerosis and microaneurysms (Figure 1) [32,33]

Lines 233-235:” Furthermore, CCXCR4 and CXCL12/SD1F, chemokines known to mediate stem cell homing, are found to be reduced in stem cells (adipose tissue-derived stem cells) from diabetic patients [64].”

Lines 269-271:” Several lines of evidence document that diabetes negatively affects CSCs homeostasis leading to insufficient replacement of old, dying cells and the acquisition by the heart a senescent phenotype [77].”

3. In lines 279-283 you wrote:” Several reports have shown that the decrease of the proliferative and differentiation potential of adult CSCs from diabetic patients is related to their senescence phenotype [81] and that the changes in chromatin conformation underlie the impaired proliferation, differentiation, and senescence of diabetic CSCs [51].”

You wrote about several reports but cited is in both cases only one paper.

Round 2

Reviewer 3 Report

Authors gave explanation of points which were mentioned in my comments and included several corresponding changes to the revised manuscript.